# Delivery of Low-Diluted Toothpaste during Brushing Improves Enamel Acid Resistance

**DOI:** 10.3390/ma16145089

**Published:** 2023-07-19

**Authors:** Ryouichi Satou, Chikara Shibata, Atsushi Takayanagi, Atsushi Yamagishi, Dowen Birkhed, Naoki Sugihara

**Affiliations:** 1Department of Epidemiology and Public Health, Tokyo Dental College, Tokyo 101-0061, Japan; shibata.chikara@wave.plala.or.jp (C.S.); tkyng@sf6.so-net.ne.jp (A.T.); camel8008@gmail.com (A.Y.); sugihara@tdc.ac.jp (N.S.); 2Fersens väg 14B, SE-211 42 Malmö, Sweden; birkhed@gmail.com

**Keywords:** toothpaste, toothpaste technique, pH-cycling, enamel, preventive dentistry

## Abstract

Toothpaste viscosity decreases rapidly when diluted with saliva during brushing, potentially causing premature washout of high-risk caries areas and reducing the uptake of dental fluoride ions. However, no reports have examined the acid resistance of enamel from the perspective of the toothpaste’s physical properties. This study aimed to elucidate the impact of toothpaste dilution on the acid resistance of the enamel, using bovine enamel as the subject. Five diluted toothpaste groups were created: a control group without toothpaste, and 100% (1.00×), 67% (1.50×), 50% (2.00×), and 25% (4.00×) dilution groups. Acid resistance was evaluated through pH cycling after toothpaste application. The results revealed a significant increase in substantial defects, compared to 67% (1.50×) at dilutions of 50% (2.00×) or higher, accompanied by a decrease in Vickers hardness. Moreover, the mineral loss increased with dilution, and a significant difference was observed between 67% (1.50×) and 50% (2.00×) (*p* < 0.01). This study revealed that the acid resistance of the enamel decreased when the dilution of toothpaste during brushing exceeded 67% (1.5×). Therefore, delivering toothpaste with a lower dilution to high-risk caries areas, including interproximal spaces and adjacent surfaces, could maintain a higher concentration of active ingredients in the toothpaste, thereby enhancing its medical effects.

## 1. Introduction

Fluoride-containing toothpaste is a beneficial source of fluoride ions in self-care for caries prevention [1]. The World Health Organization (WHO) recommends the use of fluoride-containing toothpaste for all individuals and emphasizes its significance as a crucial system for supplying fluoride worldwide [2]. The market share of fluoride-containing toothpaste is >90% in countries such as Japan and the United States. The widespread use of fluoride-containing toothpaste is considered a common factor in the reported reduction of dental caries over the past 30 years [1,3]. The caries zprevention efficacy of fluoride-containing toothpaste is high, with reported rates of 30–40% [1]. Furthermore, the application of fluoride-containing toothpaste is highly cost-effective, which is why it has garnered attention not only in developed countries, but also as a cornerstone of self-care in developing nations with a high prevalence of dental caries [4,5]. The use of fluorides has increased over recent decades, and fluoride-containing toothpastes, mouth rinses, gels, and varnishes are the modalities most widely used [5,6,7,8]. Fluoride contributes to caries prevention by promoting the crystal growth of hydroxyapatite in enamel and improving its acid resistance [3,4,5]. While many reports exist on the caries-preventing effects of medical ingredients incorporated into toothpaste, studies specifically examining the caries-preventing effects based on the physical properties of the toothpaste are lacking. Toothpaste undergoes significant changes in its physical properties owing to dilution by saliva during brushing. In particular, a negative correlation has been reported between toothpaste dilution and viscosity, highlighting the importance of toothpaste dilution as a critical factor in examining the caries-preventing effects of medical ingredients and fluoride ions [9,10].

The fluoride toothpaste technique for caries prevention has been extensively studied for 25 years (1989–2014). The Modified Fluoride Toothpaste Technique (MFTT) is well known around the world, and is also called the “2 + 2 + 2 + 2 or “Gothenburg Technique”. MFTT is a technique in which toothpaste diluted by saliva is diffused in the oral cavity as a slurry [11]. Notably, four long-term clinical studies were carried out involving children, adults, and orthodontic patients that mainly examined approximal caries, but also the buccal surface [11,12]. These studies, each spanning 2–3 years, resulted in significant caries reduction through the implementation of the MFTT. An important part of the MFTT is “post-brushing toothpaste slurry rinsing” [11], which gained attention as an interesting method at a consensus conference by 15 well-known caries experts [13]. Another approach for caries prevention is the massage method for applying fluoride toothpaste with a naked fingertip on the tooth surface, instead of using a toothbrush [14]. Both toothpaste techniques intentionally use diluted toothpaste to enhance the caries-preventing effect by maintaining high fluoride levels in the oral cavity for extended periods. In general, the higher the concentration of a medicinal ingredient such as bactericidal and hypersensitivity suppressing ingredients, the more effective it is, and medicinal ingredients in toothpaste and fluoride are no exception [15].

We hypothesized that a highly viscous, low-dilution toothpaste would provide a higher caries-preventing effect compared to a diluted slurry-like toothpaste. Furthermore, if dilution affects the efficacy while keeping the fluoride ion concentration in the toothpaste constant, it could also contribute to reducing the risk of high fluoride exposure in children, who are more susceptible to fluoride absorption and systemic circulation [5]. A decrease in toothpaste viscosity may cause premature washout of high-risk caries areas and decrease the uptake of dental fluoride ions [7]. However, no reports have examined the acid resistance of enamel from the perspective of the toothpaste’s physical properties. The purpose of this study is to elucidate the impact of toothpaste dilution on the acid resistance of the enamel.

## 2. Materials and Methods

### 2.1. Preparation of Enamel Samples and Toothpaste

A total of 45 bovine anterior mandibular teeth were used. Enamel blocks measuring 1 × 1 × 1 cm were prepared and polished to a mirror finish using water-resistant abrasive papers of #1000, #2000, and #4000 grit.

The toothpaste used in this study was Syumitect Complete ONE (GlaxoSmithKline plc., London, UK), with 1450 ppm (parts per million, approximate to mg/kg and mg/L) in the form of sodium fluoride (NaF), which is commonly available in Japan. The toothpaste was gradually diluted with ion-exchanged water to study its viscosity and decline rates. Four sample groups were prepared following the concentrations: 100% (1.00×), 67% (1.50×), 50% (2.00×), and 25% (4.00×).

### 2.2. Fluoride Toothpaste Application and pH-Cycling Acid Challenge Experiment

The samples were divided into five groups: (1) untreated fluoride toothpaste (control), (2) 100% (1.00×, undiluted), (3) 67% (1.50×), (4) 50% (2.00×), and (5) 25% (4.00×) for 2 min. Nine samples were prepared for each group (*n* = 9). In fluoride application, the toothpaste dilution solution is immersed in a 20 mL solution per tooth. To create the experimental and control surfaces on the same enamel surface, half of the mirror-polished enamel surface was covered with dental wax (Inlay Wax Soft, 27B2x00008000028, GC Co., Ltd., Tokyo, Japan). In this study, pH cycling tests were conducted in accordance with the profile shown in Figure 1. A fully automated pH-cycling system was used, which includes a control PC, a pH controller, three peristaltic pumps, and a bioreactor vessel [16]. The Stefan curve was programmed according to the method of Matsuda et al. [16,17,18]. Each cycle comprised an average of 37 ± 5 min below a pH of 5.5, followed by a recovery duration of 23 ± 3 min to restore a pH of 5.5–7.3. The total cycle duration, from the beginning to the return to the initial pH value of 7.3, averaged 60 ± 5 min. After fluoride toothpaste application for 2 min, all samples were immersed in a remineralization solution (0.02 M HEPES-based buffer solution, Ca:1.5 mM, P:0.9 mM, pH7.3, DS:5.5) for 1 h at 37 °C. After the remineralization treatment, samples underwent pH-cycling via immersion in a demineralization solution (0.2 M Lactic acid buffer solution, Ca:1.5 mM, P:0.9 mM, pH 4.5, DS:5.5) for 37 ± 5 min at 37 °C. After pH-cycling, all samples were immersed in a remineralization solution for 2 h at 37 °C. Each cycle was repeated 12 times.

### 2.3. Three-Dimensional Laser Microscopic Observation

Following wax removal, the samples underwent dehydration using an ascending ethanol series for water and ethanol displacement. To compare the step-height profile between the experimental (ES) and reference (RS) surfaces post pH-cycling, a three-dimensional (3D) measurement laser microscope (LEXT OLS4000, Olympus Corp., Tokyo, Japan) was used. The extent of tooth defects resulting from acid challenges was noted. The measurement area was 645 µm × 645 µm, with photographs taken at the boundary between the acid-demineralized ES and wax-protected RS. Finally, 3D measurements were performed at five locations for each sample, and the mean and standard deviation values were calculated. The samples were measured to determine the mean ES roughness (Sa) with a measurement area of 645 × 645 µm and a cutoff value of 80 µm. Five points on the boundary between the control and experimental surfaces were examined for the number of substantial defects and Sa per sample, and the mean ± standard deviation (SD) was computed.

### 2.4. Vickers Hardness Measurement

After the dehydration process, the Vickers hardness of the samples was measured using a hardness tester (HMV-1; Shimadzu Corp., Tokyo, Japan). The Vickers hardness (HV) values were determined by applying an indentation load of 0.49 N for 20 s. To take into account individual sample variations, the difference between the HV values before and after the experiment (∆HV = RS − ES) was calculated. The HV and ∆HV values were measured at five points on each sample, and the mean and standard deviation were calculated.

### 2.5. Cross-Sectional Morphology through Scanning Electron Microscopy

After the pH cycling, each sample was washed with xylene and then subjected to carbon vapor deposition on its surface. The tooth surfaces were examined using a scanning electron microscope (SU6600; HITACHI Ltd., Tokyo, Japan) with an acceleration voltage of 15 kV. The samples were then embedded in polyester resin (Rigolac, Nisshin EM, Tokyo, Japan) to create polished sections, which were then observed.

### 2.6. Contact Microradiography (CMR)

The imaging conditions and analysis techniques were based on those described by Angmar’s formula [19,20]. Briefly, 100 µm thick polished sections were prepared by embedding the samples in a polyester resin (Rigolac, Nisshin EM, Tokyo, Japan). Soft X-ray imaging was conducted (CMR-3, Softex, Tokyo, Japan) with a 20 µm Ni filter and light microscopy at 200× magnification using a glass plate (High Precision Photo Plate, HRP-SN-2; Konica Minolta, Tokyo, Japan). Imaging was conducted with a tube voltage of 15 kV, tube current of 3 mA, and radiation time of 15 min. The images were then analyzed using Image Pro Plus software (version 6.2; Media Cybernetics Inc., Silver Spring, MD, USA) and an image analysis system (HC-2500/OL; OLYMPUS Corp., Tokyo, Japan) to calculate the concentration profile. The extent of demineralization was evaluated by measuring the mineral loss value (ΔZ) and lesion depth (Ld). ΔZ was determined using a formula, and Ld was defined as the distance from the enamel surface to the point of the lesion with a mineral content of 95% or more in comparison to that of healthy enamel.

### 2.7. Statistical Analysis

The mean values and standard deviations were calculated for the nine samples to compare the effectiveness of the five groups. The significance of the results was determined using a Kruskal–Wallis one-way analysis of variance with a threshold value of *p* < 0.01. Post hoc comparisons were performed using the Bonferroni test. Data analysis and graphs were prepared using Origin software (ORIGIN 2023, Lightstone Corp., Tokyo, Japan). 

## 3. Results

### 3.1. Step Height Profiles Measured by 3D Laser Microscopy after pH-Cycling

Figure 2 shows the 3D laser microscopy measurements of the surface profiles after pH cycling. The left side of Figure 2a–e shows the reference surface (RS), which was not demineralized and was protected with wax, whereas the right side shows the demineralized ES. In the control group, ES was significantly demineralized, and a 35.77 ± 3.095 μm defect was observed on the enamel surface (Figure 2e,f). In the 100% (1.00×, no diluted) group, the difference in height between RS and ES decreased to 10.73 ± 2.233 μm, and a significant inhibition of demineralization was observed compared to that of the control group (*p* < 0.001) (Figure 2a,f). The 67% (1.50×) group had an even smaller height difference of 8.63 ± 3.486 μm compared to that in the 100% (1.00×) group, and the least amount of enamel demineralization was observed in the 67% (1.50×) group among the five groups (*p* < 0.001) (Figure 2b,f). The 50% (2.0×) group had 17.73 ± 2.823 μm, which was significantly more than the 100% (1.00×) and 67% (1.50×) groups (Figure 2c,f). The substantial defects of the 25% (4.00×) group were 19.05 ± 2.910 μm, similar to that of the 50% (2.00×) group and not significantly different from that of the 50% (2.00×) group (Figure 2d,f).

### 3.2. Calculated Average Roughness after pH-Cycling

Figure 3 shows a box plot of the results for each group, indicating the mean value with a gray square, the median with a horizontal line, the lower quartile as the lower limit, and the upper quartile as the upper limit. The control group exhibited significant irregularities on the enamel surface, with a mean Sa value of 0.560 ± 0.093 μm and a median of 0.545 μm (0.508–0.569). In the 100% (1.00×, no diluted) group, the mean Sa value is 0.622 ± 0.071 μm with a median of 0.632 μm (0.572–0.650). The 67% (1.50×) group exhibited the largest Sa value among the five groups, with a mean value of 0.691 ± 0.143 μm and a median of 0.723 μm (0.567–0.761); however, no significant difference was observed compared with that of the control group (*p* > 0.05). The Sa value for the 50% (2.00×) and 25% (4.00×) groups were 0.692 ± 0.088 μm with a median of 0.687 μm (0.666–0.747) and 0.689 ± 0.086 μm with a median of 0.709 μm (0.605–0.767), respectively, with no significant differences. No significant differences were observed between the groups (*p* > 0.05).

### 3.3. Micro-Vickers Hardness and Its Changes after pH-Cycling

Figure 4a shows the results of the Vickers microhardness tests for the demineralized surfaces in each experimental group. The mean Vickers hardness value of the control group was 56.27 ± 16.79, with a median of 45.40 (44.06–77.84), which was the lowest value among all groups. The Vickers hardness of the 100% (1.00×, undiluted) group had the highest value, with a mean of 214.5 ± 14.67 and a median of 208.4 (202.3–224.4), among all the groups. The Vickers hardness of the 67% (1.50×) group decreased compared to the 100% (1.00×) group, with a mean value of 147.40 ± 29.05 and a median of 156.7 (114.1–172.9). The 100% (1.00×) group exhibited significantly higher values than the control group, and a significant difference was observed between the 100% (1.00×) group and the 67% (1.50×) group (*p* < 0.01). The 50% (2.00×) group exhibited a further decrease, with a mean value of 109.3 ± 10.26 and a median of 108.7 (102.1–117.8). The 25% (4.00×) group showed a similar value, with a mean value of 108.0 ± 3.471 and a median of 107.5 (105.2–111.5), comparable to that of the 50% (2.00×) group. The variations in the values were significant between the 100% (1.00×) and 67% (1.50×) groups; however, no significant difference was observed between the 50% (2.00×) and 25% (4.00×) groups. The Vickers hardness demonstrated a decreasing trend as the dilution increased, with the 100% (1.00×) group reaching its peak hardness.

Figure 4b shows delta HV, which indicates the change in Vickers hardness before and after pH cycling. The control group exhibited the largest mean change in Vickers hardness, with a value of 266.3 ± 26.44 and a median of 260.5 (237.8–295.8). Significant differences were observed between the 100% (1.00×), 67% (1.50×), and 25% (4.00×) groups (*p* < 0.01). The 100% (1.00×) group exhibited the smallest change in Vickers hardness, with a mean value of 128.7 ± 22.80 and a median of 115.3 (108.9–153.7). Significant differences were observed between the 100% (1.00×) group and all other diluted toothpaste groups (*p* < 0.01). The mean value of the 67% (1.50×) group was 178.4 ± 31.53, with a median of 176.0 (145.7–208.7), demonstrating a significant increase compared to the 100% (1.00×) group. Furthermore, the 50% (2.00×) group had a mean value of 224.2 ± 10.15, with a median of 224.5 (219.2–230.8), indicating an increase compared with the 67% (1.50×) group, although no significant difference was observed between the 67% (1.50×) and 50% (2.00×) groups. The 25% (4.00×) group showed a similar amount of change as the 50% (2.00×) group, with a mean value of 214.0 ± 13.39 and a median value of 217.1 (201.7–226.1). The change in Vickers hardness was the smallest in the 100% (1.00×) group, and increased with dilution.

### 3.4. Cross-Sectional Scanning Electron Microscope Observations after pH-Cycling

The reflected electron image of the cross-sectioned demineralized region after pH cycling is shown in Figure 5. In the control group, a decrease in signal intensity and dissolution from the interior of enamel rods within the range of 5–10 μm below the surface were observed, resulting in a substantial reduction in the surface height due to significant loss (Figure 5a). In the 25% (4.00×) group, dissolution of enamel rods and enlargement of the enamel rod sheath were observed at a depth of 0–5 μm below the surface. Additionally, some surface depressions were observed owing to the loss of enamel rods in certain areas (Figure 5b). The 50% (2.00×) group exhibited a pattern of subsurface demineralization with dissolution of enamel rods within the range of 0–5 μm, while maintaining the surface layer. No decrease in the signal intensity was observed beyond a 10 μm depth (Figure 5c). The 67% (1.50×) group also exhibited a subsurface demineralization pattern similar to that of the 50% (2.00×) group, but with a reduced extent of demineralization (Figure 5d). In the 100% (1.00×) group, no decrease in signal intensity was observed below 5 μm. Only a localized decrease in signal intensity within the range of approximately 0–3 μm was observed (Figure 5e). When comparing the dilution of toothpaste and the extent of demineralization, a lower dilution of toothpaste corresponded to a decreased demineralization pattern (Figure 5f).

### 3.5. Measurement of Mineral Loss Value and Lesion Depth via Contact Microradiography Analysis

Figure 6 shows the mineral loss value (ΔZ, vol% μm) and lesion depth (Ld, μm) in each group using a CMR analysis. The mineral loss value in the control group was significantly higher compared to all other groups, measuring 7016 ± 1177 vol% μm (*p* < 0.01, Figure 6a). The 100% (1.00×) group exhibited the smallest mineral loss among all groups, with a decrease to 2734 ± 424.2 vol% μm, which was less than half that of the control group. The mineral loss value in the 67% (1.50×) group also exhibited a decrease to 2856 ± 341.4 vol% μm, but no significant difference was observed compared to the 100% (1.00×) group. The 50% (2.00×) group showed an increase to 3800 ± 442.5 vol% μm, surpassing the 67% (1.50×) group, but no significant difference was found between the 67% (1.50×) group and the 100% (1.00×) group. The 25% (4.00×) group displayed a significantly higher value of 4588 ± 301.1 vol% μm than the other toothpaste dilutions (*p* < 0.01).

In Ld, the Control group measured 52.19 ± 5.974 μm, the 100% (1.00×) group measured 48.30 ± 5.819 μm, the 67% (1.50×) group measured 47.61 ± 11.80 μm, the 50% (2.00×) group measured 47.28 ± 17.16 μm, and the 25% (4.00×) group measured 57.71 ± 17.29 μm. No significant differences or noticeable variations were observed among all groups, and no statistical significance was found (*p* > 0.05, Figure 6b).

## 4. Discussion

### 4.1. Comparison of Demineralization Properties by Toothpaste Dilution

The 100% (1.00×) and 67% (1.50×) groups, which had lower dilutions of toothpaste, demonstrated evident inhibition of demineralization compared to the 50% (2.00×) and 25% (4.00×) groups with higher dilutions (Figure 2, Figure 3, Figure 4, Figure 5 and Figure 6). These two groups had a significant decrease in substantial defects and a higher Vickers hardness, indicating better preservation of the tooth’s internal structure (Figure 2 and Figure 5). Furthermore, the two groups with lower dilutions of toothpaste exhibited a significant reduction in mineral loss compared to the 25% (4.00×) group, indicating quantitative inhibition of enamel demineralization (Figure 6a). Significant differences were observed between the 67% (1.50×) and 50% (2.00×) groups in terms of substantial defects, Vickers hardness, and mineral loss. No significant differences were observed between the 50% (2.00×) and 25% (4.00×) groups in terms of substantial defects, Vickers hardness, and mineral loss, suggesting that no significant difference existed when the toothpaste was diluted to 50% (2.00×) or higher (Figure 2, Figure 4 and Figure 6a). These results are consistent with those of previous studies, which have shown that the uptake of fluoride by hydroxyapatite significantly decreases when toothpaste is diluted to a ratio of 1.75 or higher. The reduction in fluoride ions’ action on enamel may explain the substantial increase in enamel loss and the decrease in Vickers hardness [6,7,21,22]. Previous studies have also found no significant differences in fluoride ion uptake between the 2.00×, 3.00×, 4.00×, and 5.00× groups, which aligns with the results of this study. No difference exists between the 50% (2.00×) and 25% (4.00×) groups [6].

The SEM images for each dilution ratio in this study provide supporting evidence for the Vickers hardness results. The 100% (1.00×) and 67% (1.50×) groups, with lower dilutions, exhibited less structural breakdown of tooth enamel than the 50% (2.00×) group. A decrease in enamel rod demineralization and an improvement in surface demineralization patterns were shown in the low-dilution group (Figure 5). It has been reported that fluoride-containing toothpaste supplies high concentrations of fluoride ions to the surface of the enamel and, through reactions with calcium ions present in the saliva and on the surface of the tooth, forms calcium-like fluoride particles [1,4,5,21]. When the pH of the tooth surface decreases to 4–5 due to food and beverage consumption, the fluoride-like calcium particles dissolve, creating a supersaturated state in which the fluoride and calcium ion concentrations in the microenvironment rapidly increase. This has been shown to enhance acid resistance [1,5,21]. Furthermore, it has been observed that the quantity and particle size of the formed calcium fluoride (CaF_2_) on the tooth surface increase with higher concentrations of fluoride ions in toothpaste, thereby providing higher acid resistance [1,15]. The fluoride ion concentration of the toothpaste used in this study was 1450 ppm. Based on the calculations, it can be assumed that the 100% (1.00×) group had a fluoride ion concentration of 1450 ppm F, the 67% (1.50×) group had 1005 ppm F, the 50% (2.00×) group had 750 ppm F, and the 25% (4.00×) group had 375 ppm F, which would react with the enamel. Indeed, in reality, factors such as changes in toothpaste viscosity due to dilution and interactions with toothpaste ingredients prevent a complete reaction with the entire quantity of toothpaste. Therefore, it is likely that the actual interactions occurred at lower concentrations than those previously mentioned. The lowest fluoride ion concentration at which CaF_2_ is formed at neutral pH is 300 ppm F [23]. Koga et al. examined the amount of fluoride uptake into the enamel as a function of brushing time and toothpaste usage. They reported that at a reaction time of 120 s, no significant difference exists between enamel surface F^−^ concentrations of 300 and 500 ppm, and that fluoride uptake was significantly higher only above 1000 ppm F [24]. In this study, enamel interaction at concentrations exceeding 1000 ppm was possible in the 100% (1.00×) and 67% (1.50×) groups. Our results also demonstrated a substantial increase in enamel loss and mineral loss with dilutions of 50% (2.00×) or higher (Figure 2 and Figure 6). Therefore, it is suggested that the 100% (1.00×) and 67% (1.50×) groups with toothpaste dilutions lower than 50% (2.00×) enhanced enamel acid resistance by increasing fluoride’s incorporation into the tooth structure and generating a significant amount of CaF_2_ on the tooth surface.

### 4.2. Evaluation Methods, Limitations, and Their Application to Dental Clinical Practice

In this study, instead of the conventional method of immersing the samples in a demineralizing solution for a fixed period, pH cycling was adopted as the acid challenge method. This choice was made because of the dynamic nature of caries progression, which involves fluctuations between the demineralization and remineralization processes. By replicating pH fluctuations that resemble the Stephan curve, pH cycling can simulate a more realistic oral environment than solution immersion [25]. Many previous in vitro studies have investigated demineralization and remineralization using a pH cycling model and toothpaste, similar to the present study. Itthagarun et al. conducted pH cycling on primary enamel after the application of non-fluoride-containing toothpaste (0 ppm fluoride) and toothpaste containing 500 ppm fluoride. They observed a significant reduction in demineralization progression in the group that used toothpaste containing 500 ppm F fluoride [26,27]. Toda et al. applied a placebo toothpaste (0 ppm F) along with toothpastes containing 250 ppm F and 1000 ppm F, and reported that higher fluoride concentrations resulted in greater inhibition of demineralization through pH cycling [28]. These findings align with the results of the present study, indicating that higher fluoride ion concentrations applied to the tooth surface lead to improved acid resistance. Previous studies have focused on directly manipulating the fluoride concentration in toothpaste to investigate enamel acid resistance. However, the present study adds a new perspective by considering changes in the concentration and physical properties of toothpaste through dilution. The results of this study suggest that when evaluating the effectiveness of the medicinal components of toothpaste, it is important to consider not only the replication of the oral environment through pH cycling, but also the influence of dilution by saliva. There are significant individual variations in salivary flow rate and composition, and a decrease in saliva secretion, particularly among the elderly (due to medication use), has been widely reported [29,30]. In this study, the demineralization–remineralization solution had a high saturation of calcium and phosphate, resulting in a substantial defect of approximately 35 μm in the control group (Figure 2). However, it is important to note that under conditions of dry mouth or reduced salivary remineralization capacity, the acid resistance of the enamel may further decrease beyond the findings of this study.

The limitations of this study include several factors. First, all dilution magnification groups were subjected to 2 min tooth immersion; therefore, differences in salivary flow, brushing behavior, and the duration of action due to washout were not replicated. However, previous research reported that dilutions of 57% (1.75×) or higher result in a decrease in toothpaste viscosity, leading to >50% washout of the toothpaste on interproximal and adjacent surfaces during a 2 min brushing time [7]. Therefore, it is suggested that the enamel acid resistance at dilution ratios of 50% (2.00×) or higher would further decrease. Second, the only type of fluoride used in this study is sodium fluoride. Different fluoride compounds, such as sodium monofluorophosphate and stannous fluoride, which have different mechanisms of action, may yield different results. Furthermore, it would be desirable to conduct additional evaluations of enamel acid resistance in vivo, owing to the inhibitory effects of salivary proteins, enzymes, and the pellicle, that which on demineralization, as well as the absence of bacteria in in vitro experiments. Bovine tooth enamel, which unlike human enamel has no exposure to fluoride, was used in this study. Although bovine teeth are commonly used for in vitro experiments in cariology because of their low individual variability, future experiments should be conducted in the human oral cavity. Moreover, our research focused on the assessment of enamel acid resistance, with the aspect of remineralization currently unknown. Kawasaki et al. conducted an experiment in which suspensions of fluoride-containing toothpaste at concentrations of 0, 500, 1000, and 4000 ppm were immersed in initial enamel demineralization samples three times a day for a duration of five minutes. The extent of remineralization was assessed using the QLF method. The results indicated that the degree of remineralization by the toothpaste suspensions was low for the 0 ppm F and 4000 ppm F groups, while it was high for the 500 ppm F and 1000 ppm F groups. Additionally, they reported no significant differences in the degree of remineralization between the 500 and 1000 ppm F groups [25]. Cate et al. investigated the impact of fluoride-containing toothpaste at concentrations ranging from 0 to 3000 ppm on remineralization in mild and severe initial caries using a pH-cycling model. They reported only a slight difference in remineralization between toothpaste containing 1000 ppm F and that containing 3000 ppm F [31]. Furthermore, a previous study reported that the effectiveness of fluoride in severe initial caries occurs at concentrations up to 1250 ppm [32]. Previous studies have indicated a relationship between the concentration of fluoride ions that act on demineralized enamel and remineralization. It is also anticipated that a decrease in fluoride ion concentration due to dilution would similarly affect the process of remineralization. In this study, we used a fluoride concentration of 1450 ppm, which is commonly accessible in Japan and is regulated by pharmaceutical laws. However, in countries such as the United States and Sweden, high-concentration toothpaste containing 5000 ppm F is available for purchase without a prescription. Toothpaste containing 5000 ppm F is reported to generate a significant amount of fluoride-like calcium substances on the tooth surface when it reacts with the tooth surface, thereby improving acid resistance. However, it inhibits mineral diffusion into the subsurface demineralized area, thereby suppressing the remineralization process [31]. An inhibitory effect on remineralization has also been reported for toothpaste containing 4000 ppm of F [20,31]. Therefore, when using toothpaste with 5000 ppm F, inhibiting remineralization at a dilution range of 1.00×–1.50× is possible, which we recommend. It is necessary to experimentally determine an appropriate dilution ratio to ensure optimal results. The results of this study can be applied to other dentifrices containing sodium fluoride, because the mechanism of action is the same. Our theory focuses on toothpaste with fluoride concentrations of up to 1450 ppm, and individual verification is deemed necessary for toothpastes with higher fluoride concentrations.

The dental findings of this study offer evidence-based guidance for patient brushing, implying that the use of a small amount of toothpaste can result in a dilution of over 50% (2.00×) immediately after the start of brushing, indicating limited expectations for improving enamel acid resistance. Therefore, this study serves as a basis for recommending appropriate and sufficient use of toothpaste during brushing to prevent dilution. Compared to professional care, which is performed only a few times a year, self-care provides by far the most opportunities to react on the teeth. Therefore, the effects of changing toothpaste and brushing technique on caries prevention are significant. Additionally, they provide evidence for the effectiveness of toothpaste techniques that target low-dilution, high-risk caries areas and minimize early washout caused by dilution, thereby contributing to caries prevention.

## 5. Conclusions

In this study, we qualitatively and quantitatively evaluated enamel acid resistance after the application of each diluted toothpaste on bovine tooth enamel. This study revealed that applying toothpaste with dilutions lower than 67% (1.50×) to the tooth surface significantly improved the enamel’s acid resistance. Acid resistance among dilution groups beyond 50% (2.00×) showed no significant qualitative or quantitative differences. Delivering toothpaste with a lower dilution to high-risk caries areas such as interproximal spaces and adjacent surfaces could maintain a higher concentration of active ingredients in the toothpaste, thereby enhancing its medical effects. It is suggested that when evaluating toothpaste and acid resistance, it is necessary to consider not only the concentration of medicinal ingredients, but also the changes in properties such as dilution and viscosity during a 2 min brushing. Changes in toothpaste and brushing technique, which are parts of routine self-care, can contribute to improved effectiveness in the prevention of caries.

## Figures and Tables

**Figure 1 materials-16-05089-f001:**
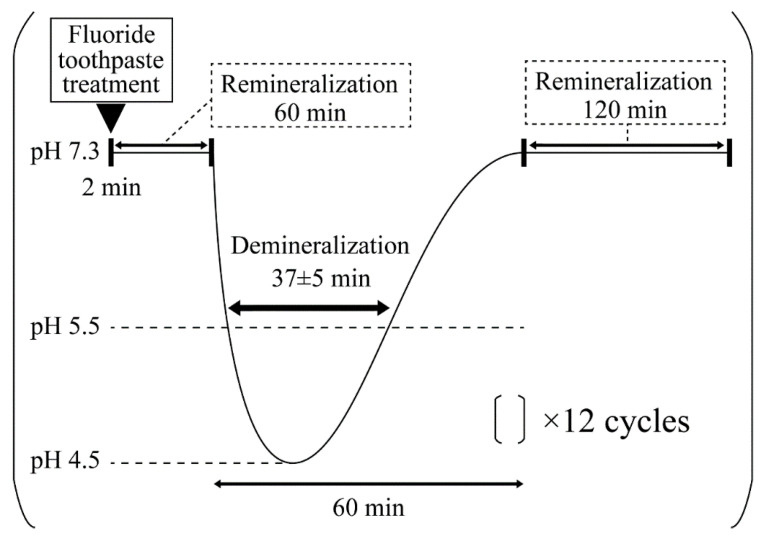
Overview of the fluoride toothpaste application and pH-cycling acid challenge. Overview of the programmed temporal pH changes in the pH-cycling system. The time below the critical pH of the enamel (pH 5.5) is considered the demineralization time and is set to 37 ± 5 min.

**Figure 2 materials-16-05089-f002:**
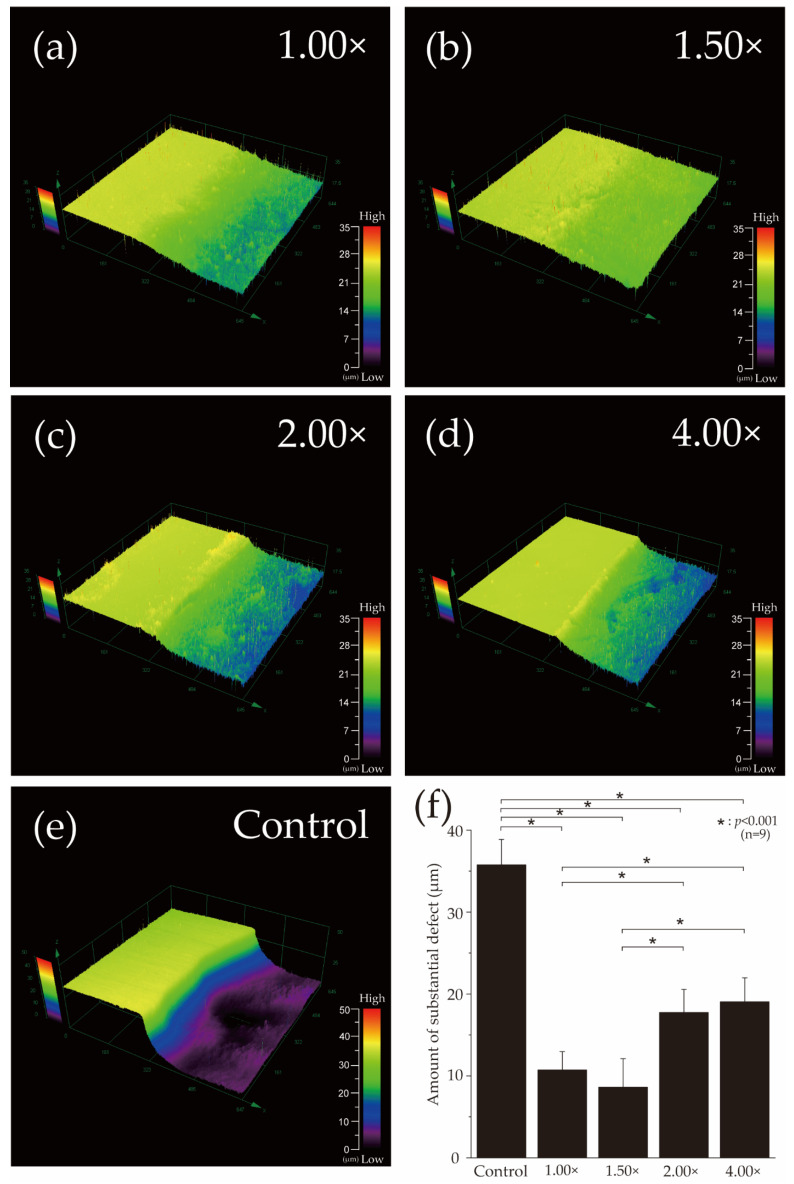
Height difference profiles measured using a 3D laser microscope. Boundary images of the reference and experimental surfaces after pH-cycling of the (**a**) 100% (1.00×, no diluted), (**b**) 67% (1.50×), (**c**) 50% (2.00×), (**d**) 25% (4.00×), and (**e**) control groups. The left side of (**a**–**e**) shows the RS protected by wax and not demineralized. The right side shows the ES that has been demineralized. (**f**) Graphical representation of the substantial defects due to demineralization (*n* = among the three groups (*n* = 9, * *p* < 0.001).

**Figure 3 materials-16-05089-f003:**
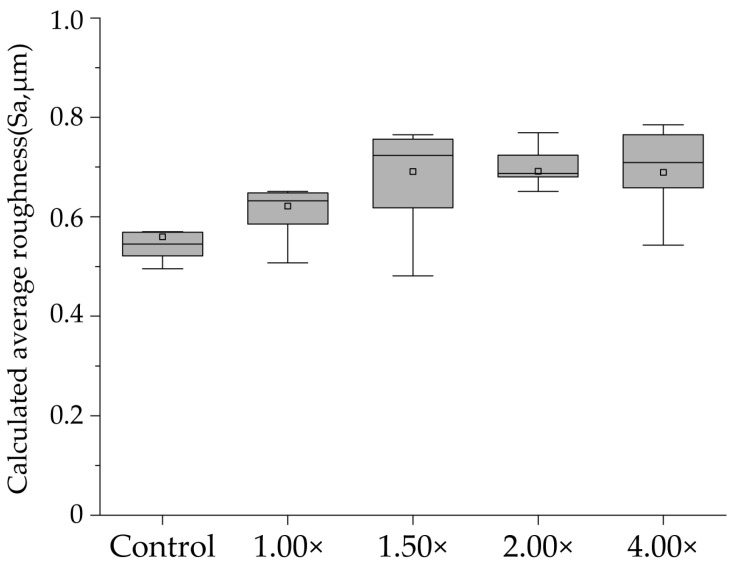
Calculated average roughness after pH-cycling. The distribution of the dataset can be visualized using the boxplot (*n* = 9). The median values are indicated by a horizontal line in the middle of the box, and the lower and upper boundaries indicate the 25th and 75th percentiles, respectively. Gray squares indicate mean values.

**Figure 4 materials-16-05089-f004:**
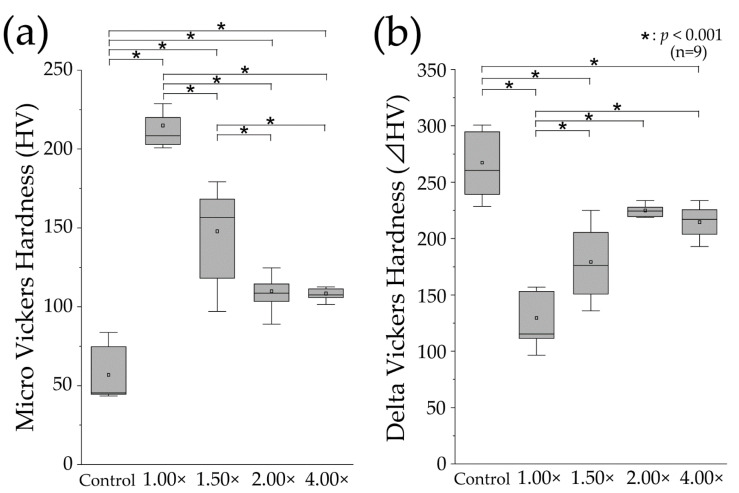
Micro-Vickers hardness measurements. (**a**) Boxplot of micro-Vickers hardness (HV) values after pH-cycling (*n* = 9, * *p* < 0.001). The median value is indicated by the horizontal line in the middle of the box, and the lower and upper boundaries indicate the 25th and 75th percentiles, respectively. The white squares indicate the mean value. (**b**) Boxplot of ΔHV values (difference in the HV values between the RS and ES) (*n* = 9, * *p* < 0.001).

**Figure 5 materials-16-05089-f005:**
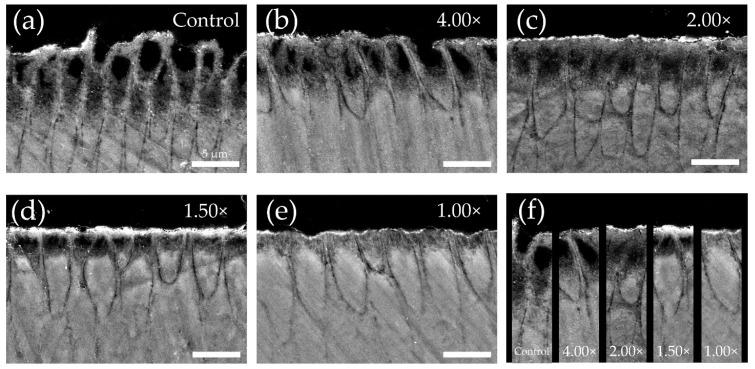
Scanning electron microscope (SEM) images of enamel cross-sections after pH-cycling. Cross-sectional SEM image of the control (**a**), 25% (4.00×) (**b**), 50% (2.00×) (**c**), 67% (1.50×) (**d**) and 100% (1.00×) groups (**e**). (**f**) Comparative images of each dilution magnification. (**a**–**f**) Scale bar is 5 μm. All images were recorded at 5000-fold magnification with a carbon deposition sample.

**Figure 6 materials-16-05089-f006:**
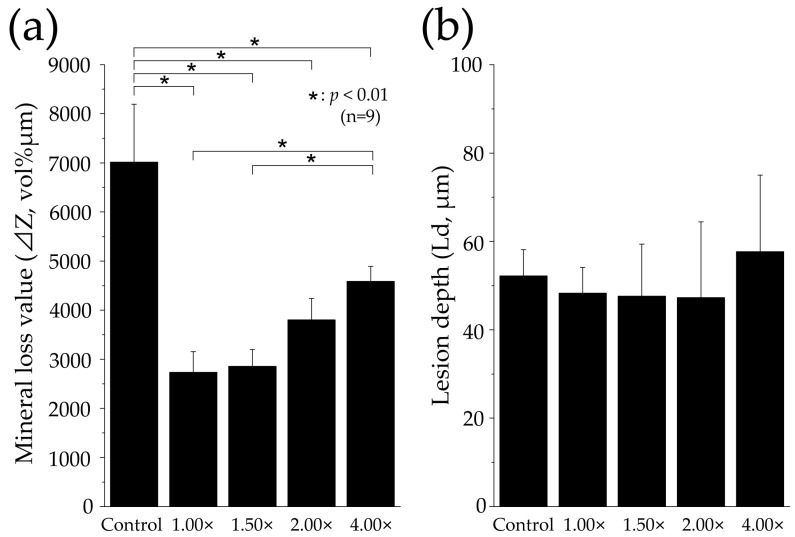
Graph representation of mineral loss (ΔZ) and lesion depth (Ld) values after pH-cycling. (**a**) Graphical representation of mineral loss value (ΔZ, *n* = 9, * *p* < 0.01). All nine samples were measured, and the mean ± SD is shown. (**b**) Graphical representation of lesion depth value (Ld, *n* = 9). From the surface prior to the demineralization experiment, the depth of demineralization was determined up to a site showing 95% healthy enamel. All nine samples are measured, and the mean ± SD is shown.

## Data Availability

All data are included in the manuscript.

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
