# Peer review of "Delivery of Low-Diluted Toothpaste during Brushing Improves Enamel Acid Resistance"

_materials, 2023, doi:10.3390/ma16145089_

Round 1

Reviewer 1 Report

Overall, the manuscript appears to be well-structured and addresses an important topic related to the impact of toothpaste dilution on enamel acid resistance. Here are some comments on the different sections:

Title: The title is clear and concise, effectively conveying the main focus of the study.

Abstract:

  1. It would be helpful to mention the specific reasons why enamel acid resistance is important, such as its role in preventing dental caries or tooth decay.
  2. Consider including the main findings of the study in the abstract, as this will provide readers with a clear overview of the results.

Keywords: The selected keywords are relevant and appropriate for the study. They cover the main aspects of the research.

Results and discussion have been written scientifically

General Suggestions:

  1. Provide a brief introduction section at the beginning of the manuscript to provide background information on enamel acid resistance, toothpaste formulation, and the importance of fluoride in dental care.
  2. In the abstract and conclusion, consider highlighting the practical implications of the findings and how they can be applied to improve dental care practices.
  3. Throughout the manuscript, consider providing a more detailed description of the pH cycling method used to evaluate acid resistance, as it is a critical aspect of the study.
  4. Add more information about the potential limitations of the study, such as the use of bovine enamel instead of human enamel and the need for further research to validate the findings in clinical settings.

Conclusion:

  1. The conclusions accurately summarize the findings of the study.
  2. Consider providing a brief summary of the methodology used in the research, as this will help readers understand the context of the conclusions.
  3. It would be beneficial to discuss the potential implications of the study's findings for dental practice or public health.
  4. Suggest mentioning any limitations or potential areas for future research to provide a comprehensive conclusion.

Minor editing of English language required

Author Response

Overall, the manuscript appears to be well-structured and addresses an important topic related to the impact of toothpaste dilution on enamel acid resistance. Here are some comments on the different sections:

> We strongly appreciate the reviewer's comment. We are thankful for the time and energy you expended.

Title: The title is clear and concise, effectively conveying the main focus of the study.

Abstract:

It would be helpful to mention the specific reasons why enamel acid resistance is important, such as its role in preventing dental caries or tooth decay.

Consider including the main findings of the study in the abstract, as this will provide readers with a clear overview of the results.

> The reviewer's comment is correct. In accordance with the reviewer's comment, we have added introduction part as described below.

Page 1, Line 18-20

Moreover, the mineral loss increased with dilution, and a significant difference was observed between 67% (1.50×) and 50% (2.00×) (p<0.01).

Page 1, Line 40-42

Fluoride contributes to caries prevention by promoting the crystal growth of hydroxy-apatite in enamel and improving its acid resistance [3-5].

Keywords: The selected keywords are relevant and appropriate for the study. They cover the main aspects of the research.

Results and discussion have been written scientifically

> We appreciate the reviewer's comment on this point.

General Suggestions:

Provide a brief introduction section at the beginning of the manuscript to provide background information on enamel acid resistance, toothpaste formulation, and the importance of fluoride in dental care.

In the abstract and conclusion, consider highlighting the practical implications of the findings and how they can be applied to improve dental care practices.

Throughout the manuscript, consider providing a more detailed description of the pH cycling method used to evaluate acid resistance, as it is a critical aspect of the study.

Add more information about the potential limitations of the study, such as the use of bovine enamel instead of human enamel and the need for further research to validate the findings in clinical settings.

> In accordance with the reviewer's comment, we have added each part as described below.

<Introduction>

Page 1, Line 38-40

The use of fluorides has increased over recent decades and fluoride containing toothpastes, mouthrinses, gels and varnishes, are the modalities most widely used [5].

Page 1, Line 40-42

Fluoride contributes to caries prevention by promoting the crystal growth of hydroxy-apatite in enamel and improving its acid resistance [3-5].

<Material and method>

Page 2, Line 95-96

Fully automated pH-cycling system was used, which includes a control PC, a pH controller, three peristaltic pumps, and a bioreactor vessel [16].

<Discussion>

Page 11, Line 385-388

Bovine tooth enamel, which unlike human enamel has no exposure to fluoride, was used in this study. Although bovine teeth are commonly used for in vitro experiments in cariology because of their low individual variability, future experiments should be conducted in the human oral cavity.

Page 12, Line 426-429

Compared to professional care, which is performed only a few times a year, self-care provides by far the most opportunities to react on the teeth. Therefore, the effect on caries prevention of changing toothpaste and brushing technique is significant.

Conclusion:

The conclusions accurately summarize the findings of the study.

Consider providing a brief summary of the methodology used in the research, as this will help readers understand the context of the conclusions.

It would be beneficial to discuss the potential implications of the study's findings for dental practice or public health.

Suggest mentioning any limitations or potential areas for future research to provide a comprehensive conclusion.

> The reviewer's comment is correct. In accordance with the reviewer's comment, we have added introduction part as described below.

<Conclusion>

Page 12, Line 434-435

In this study, we qualitatively and quantitatively evaluated enamel acid resistance after the application of each diluted toothpaste on bovine tooth enamel.

Page 12, Line 444-445

Changes in toothpaste and brushing technique, which are routine self-care, can contribute to improved caries-preventive effectiveness.

Again, thank you for giving us the opportunity to strengthen our manuscript with your valuable comments and queries. We have worked hard to incorporate your feedback and hope that these revisions persuade you to accept our submission.

Reviewer 2 Report

It is opinion of the reviewer that this paper before acceptance needs several corrections. My individual comments are listed below.

The title should be written with first capital letters.

L. 6/7 – The authors initial and their e-mail addresses should be added.

L. 47 – A “Gothenburg Technique” should be a bit described.

L. 58 – Some medicinal ingredients should be mentioned.

L. 63 – The risk should be confirmed by a reference citation.

L. 76 and entire paper– ppm is not any SI unit, correction is needed.

L. 102 – The dehydration should be briefly described.

L. 154 and entire paper – The results > 10 should be reported with two digitals after decimal point, >100 with one, >1000 without any digitals.

L. 456 -  It should be “PLoS ONE”

L. 506 – It should be “pH”.

L. 508 – “in vitro” with italic.

Correction by a native speaker is not needed.

Author Response

It is opinion of the reviewer that this paper before acceptance needs several corrections. My individual comments are listed below.

> We strongly appreciate the reviewer's comment. We are thankful for the time and energy you expended.

The title should be written with first capital letters.

> The reviewer's comment is correct. We fixed the title.

Page 1, title

Delivery of Low-Diluted Toothpaste During Brushing Improves Enamel Acid Resistance

  1. 6/7 – The authors initial and their e-mail addresses should be added.

> The reviewer's comment is correct. We added the authors initial and their e-mail addresses.

Page1, Line 6-7

1 Department of Epidemiology and Public Health, Tokyo Dental College, Tokyo 101-0061, Japan; sugiha-ra@tdc.ac.jp (N.S.)

2 Fersens väg 14B, SE-211 42, Malmö, Sweden; birkhed@gmail.com (DB)

  1. 47 – A “Gothenburg Technique” should be a bit described.

> We appreciate the reviewer's comment on this point. In accordance with the reviewer's comment, we have added introduction part as described below.

Page 2, Line 52-53

MFTT is a technique in which toothpaste diluted by saliva is diffused in the oral cavity as a slurry [11].

  1. 58 – Some medicinal ingredients should be mentioned.

> In accordance with the reviewer's comment, we have added introduction part as described below.

Page 2, Line 63-65

In general, the higher the concentration of a medicinal ingredient such as bactericidal and hypersensitivity suppressing ingredients, the more effective it is, and medicinal ingredients in toothpaste and fluoride are no exception [15].

  1. 63 – The risk should be confirmed by a reference citation.

> In accordance with the reviewer's comment, we have added reference as described below.

Page 2, Line 67-70

Furthermore, if dilution affects the efficacy while keeping the fluoride ion concentration in the toothpaste constant, it could also contribute to reducing the risk of high fluoride exposure in children, who are more susceptible to fluoride absorption and systemic circulation [5].

  1. 76 and entire paper– ppm is not any SI unit, correction is needed.

> In accordance with the reviewer's comment, we have added the following sentence.(The meaning of the SI units mg/kg and mg/L was approximated by ppm.) In dentistry, it is standard practice to express fluoride concentrations in ppm, and textbooks and technical books in dental schools also use ppm. Since the readers of this paper are likely to be dental professionals, we would like to use ppm.

Page 2, Line 81-83

The toothpaste used in this study was Syumitect Complete ONE (GlaxoSmithKline plc., London, England), with 1500 ppm (parts per million, approximate to mg/kg and mg/L) in the form of sodium fluoride (NaF),

  1. 102 – The dehydration should be briefly described.

> In accordance with the reviewer's comment, we have added materials and methods part as described below.

Page 3, Line 112-113

Following wax removal, the samples underwent dehydration using an ascending ethanol series for water and ethanol displacement.

  1. 154 and entire paper – The results > 10 should be reported with two digitals after decimal point, >100 with one, >1000 without any digitals.

> We appreciate the reviewer's comment on this point. In accordance with the reviewer's comment, we have fixed all number representation.

  1. 456 - It should be “PLoS ONE”

> In accordance with the reviewer's comment, we have fixed reference No.10.

  1. 506 – It should be “pH”.

> We appreciate the reviewer's comment. In accordance with the reviewer's comment, we have fixed reference No.26.

  1. 508 – “in vitro” with italic.

> In accordance with the reviewer's comment, we have fixed reference No.27.

Again, thank you for giving us the opportunity to strengthen our manuscript with your valuable comments and queries. We have worked hard to incorporate your feedback and hope that these revisions persuade you to accept our submission.

Reviewer 3 Report

The manuscript is interesting and a definite contribution. However, some issues must be addressed to make the manuscript better.

1. Why use "Syumitect Complete ONE (GlaxoSmithKline 75 plc., London, England)" toothpaste in particular?

2."Fluoride toothpaste application and pH-cycling acid challenge experiment" only mentions the ratio of toothpaste. However, how much was actually used was not presented. Also, why is n=9?

3. I don't understand the author is it just soaking or brushing with a toothbrush?

4. Why does the control group have different results for a and b in Figure 4.?

5. What are "high-risk areas"?

6. The purpose of this study should be emphasized in the Introduction.

7. The manuscript uses only one toothpaste for experiments. Are the conclusions of the experiment applicable to other toothpastes?

8. The environment of the oral cavity is complex, can the teeth used in the manuscript represent human teeth?

Minor editing of English language required

Author Response

The manuscript is interesting and a definite contribution. However, some issues must be addressed to make the manuscript better.

> We strongly appreciate the reviewer's comment. We are thankful for the time and energy you expended.

  1. Why use "Syumitect Complete ONE (GlaxoSmithKline 75 plc., London, England)" toothpaste in particular?

> This toothpaste is sold worldwide and has a large market share. In addition, the reasons for selecting it are that it does not require a prescription and is easily obtainable, and it contains sodium fluoride, which is the most commonly used ingredient in toothpaste, with a fluoride concentration of 1450 ppm. We thought this toothpaste would be a good model to generalize the results of this study.

2."Fluoride toothpaste application and pH-cycling acid challenge experiment" only mentions the ratio of toothpaste. However, how much was actually used was not presented.

>The reviewer's comment is correct. This experiment is prepared and dipped in a dilute solution of toothpaste in a volume of 20 ml per one tooth. The following text was added to the

Page 2, Line 90-91

In fluoride application, the toothpaste dilution solution is immersed in a 20ml solution per tooth.

Also, why is n=9?

>We found that similar previous studies have been performed with n=5 to 15 (CMR and height difference profiles measured using a 3D laser microscope). We started with n=12 samples per group. However, handling 100μm thick samples for CMR was difficult and 1-3 samples in each group were broken. Therefore, the final number of samples was n=9.

  1. I don't understand the author is it just soaking or brushing with a toothbrush?

> I'm sorry for the lack of clarity. It just soaking is correct. The following text has been added for clarity.

Page 2, Line 90-91

In fluoride application, the toothpaste dilution solution is immersed in a 20ml solution per tooth.

  1. Why does the control group have different results for a and b in Figure 4.?

> HV is the measured value, but ∆HV is the amount of change (∆HV = RS - ES), so the control group values are different. Please refer to the following section.

Page 3-4, Line 127-132

To take into account individual sample variations, the difference between the HV values before and after the experiment (∆HV = RS - ES) was calculated. The HV and ∆HV values were measured at five points on each sample and the mean and standard deviation were calculated.

  1. What are "high-risk areas"?

> High-risk areas are areas of the oral cavity that are at high risk for caries. Specifically, it refers to interproximal spaces and adjacent surfaces of the teeth. The text in the Conclusion has been modified for clarity.

Page12, Line 439-441

Delivering toothpaste with a lower dilution to high-risk caries areas such as inter-proximal spaces and adjacent surfaces could maintain a higher concentration of active ingredients in the toothpaste, thereby enhancing its medical effects.

  1. The purpose of this study should be emphasized in the Introduction.

> In accordance with the reviewer's comment, we have modified the following sentence.

Page 2, Line 74-75

The purpose of this study is to elucidate the impact of toothpaste dilution on the acid resistance of the enamel.

  1. The manuscript uses only one toothpaste for experiments. Are the conclusions of the experiment applicable to other toothpastes?

>If the type of fluoride in the toothpaste is sodium fluoride, it may be applicable to others since the mechanism of action is the same; it may not be applicable to toothpastes containing MFP or sodium fluoride concentration different from 1450ppm. The following text was added to the discussion limitation.

Page 12, Line 417-419

The results of this study can be applied to other dentifrices containing sodium fluoride because the mechanism of action is the same.

  1. The environment of the oral cavity is complex, can the teeth used in the manuscript represent human teeth?

> Bovine teeth were used in this study. Bovine teeth are less individualized and more common in dental in vitro experiments than human teeth because bovine teeth are not fluoridated. Experiments using human teeth are described in the discussion Prospects for Clinical Dentistry.

Page 11, Line 385-388

Bovine tooth enamel, which unlike human enamel has no exposure to fluoride, was used in this study. Although bovine teeth are commonly used for in vitro experiments in cariology because of their low individual variability, future experiments should be conducted in the human oral cavity.

Again, thank you for giving us the opportunity to strengthen our manuscript with your valuable comments and queries. We have worked hard to incorporate your feedback and hope that these revisions persuade you to accept our submission.

Round 2

Reviewer 3 Report

Thanks for the author's reply.

The manuscript has been well improved. I think it's acceptable for publication.

Minor editing of English language required